# Effects of Plant-Growth-Promoting Rhizobacteria (PGPR) and Cyanobacteria on Botanical Characteristics of Tomato (*Solanum lycopersicon* L.) Plants

**DOI:** 10.3390/plants11202732

**Published:** 2022-10-15

**Authors:** Ebtesam A. Gashash, Nahid A. Osman, Abdulaziz A. Alsahli, Heba M. Hewait, Ashmawi E. Ashmawi, Khalid S. Alshallash, Ahmed M. El-Taher, Enas S. Azab, Hany S. Abd El-Raouf, Mohamed F. M. Ibrahim

**Affiliations:** 1Department of Chemistry, Faculty of Arts and Science in Baljurashi, Baha University, Baha 65635, Saudi Arabia; 2Department of Science and Technology, Ranya Collage, Taif University, Taif 21944, Saudi Arabia; 3Department of Botany and Microbiology, Science College, King Saud University, P.O. Box 2455, Riyadh 11451, Saudi Arabia; 4Soils & Water and Environment Research Institute, Agricultural Research Center, Giza 12112, Egypt; 5Department of Horticulture, Faculty of Agriculture, Al-Azhar University, Cairo 11651, Egypt; 6College of Science and Humanities-Huraymila, Imam Mohammed Bin Saud Islamic University (IMSIU), Riyadh 11432, Saudi Arabia; 7Department of Agricultural Botany, Agriculture Faculty, Al-Azhar University, Cairo 11651, Egypt; 8Agricultural Botany Department, Faculty of Agriculture, Suez Canal University, Ismailia 41522, Egypt; 9Department of Biology, University College, Taif University, Taif 21944, Saudi Arabia; 10Department of Agricultural Botany, Faculty of Agriculture, Ain Shams University, Cairo 11566, Egypt

**Keywords:** biofertilizers, *Bacillus subtilis*, *Bacillus amyloliquefaciens*, total soluble solids (TSS), ascorbic acid, anatomy

## Abstract

Tomatoes are an important agricultural product because they contain high concentrations of bioactive substances, such as folate, ascorbate, polyphenols, and carotenoids, as well as many other essential elements. As a result, tomatoes are thought to be extremely beneficial to human health. Chemical fertilizers and insecticides are routinely utilized to maximize tomato production. In this context, microbial inoculations, particularly those containing PGPR, may be utilized in place of chemical fertilizers and pesticides. In this study, we investigated the effects of PGPR (*Bacillus subtilis*, and *Bacillus amyloliquefaciens*) and cyanobacteria when utilized alone, and in conjunction with each other, on the growth, quality, and yield of fresh fruits of tomato plants. The results showed that the inoculation significantly increased all measured parameters of tomato plants compared with the control. Combined use of *B. subtilis* and *B. amyloliquefaciens* had a positive impact on tomato yield, increasing fruit yield. Moreover, leaflet anatomical characteristics were altered, with increased thickness of the upper epidermis, lower epidermis, palisade tissue, spongy tissue, and vascular bundles. Tomato fruit quality was improved, as measured by an increased number of fruit per plant (76% increase), fruit weight (g; 33% increase), fruit height (cm; 50% increase), fruit diameter (cm; 50%), total soluble solids (TSS; 26% increase), and ascorbic acid (mg/100 g F.W.; 75% increase), relative to the control, in the first season. In addition, fruit chemical contents (N, P, and K) were increased with inoculation. The results suggest that inoculation with *B. subtilis* and *B. amyloliquefaciens* could be successfully used to enhance tomato plant growth and yield.

## 1. Introduction

The great economic value of tomatoes makes them a promising horticultural crop for cultivation [1]. Next to potatoes, tomatoes (*S. lycopersicon*) are considered to be the second most important vegetable crop, in terms of agricultural consequences for human consumption [2]. Since tomatoes are a significant source of vitamins A and C in many Western nations, including the United States, a higher per-person intake could be beneficial [3]. Increasing the amounts of major phytochemicals with strong health advantages found in tomatoes, such as carotenoids, glycoalkaloids, ascorbic acid, tocopherols, and other phenolic and flavonoid components, is another possibility [4].

Designing workable biological approaches, and utilizing microbial resources that can support plant growth and soil health, are the main tactics used for crop production in sustainable agriculture [5]. The study of plant–microbe interactions, and how they affect ecological niches, is expanding. However, there is still room for improvement in our understanding of the interactions between rhizosphere organisms for improved plant development and sustainable agriculture. Numerous microorganisms that are naturally present in soil, such as bacteria, fungus, and archaea, have been isolated and identified as having the potential to be delivered in agricultural systems [6,7]. A microhabitat’s microbial population is influenced by both biotic (plants and other species) and abiotic (environment and management techniques) elements [8]. Plant development and crop yield have benefited from the recruitment of microbes from environmental samples [9]. There are tens of thousands of different kinds of bacteria that are connected to plant roots, making this a very diverse group. The intricate microbial community associated with a plant is also known as the plant’s second genome, and is extremely important for plant health. Since various plant species host diverse microbiological communities when cultivated in the same soil, recent developments in the study of plant–microbe interactions have shown that plants can modify their rhizosphere microbiomes [10].

According to reports, the microbial population is typically higher in the soil near plant roots (rhizosphere) than in most other soil because plant roots exudate secretions containing secondary metabolites, such as sugar and amino acids, which act as the microorganisms’ energy source. The health, growth, yield, and quality of the crop harvest are all influenced by the interactions between plants and their rhizosphere microbial communities [6]. Recently, interest has grown in the use of beneficial microorganisms in soilless cultures to induce plant resistance to biotic and abiotic stress factors, and to increase plant growth and yield [11,12].

Many microbes have the capacity to promote plant growth, and microbial products that enhance plant health and growth have been commercialized. These types of bacteria have been designated plant growth-promoting rhizobacteria [13]. The significant beneficial effects of these rhizobacteria on plant growth are achieved through both direct and indirect mechanisms. A range of mechanisms, such as biological nitrogen fixation, ethylene level reduction, siderophore production, phytohormone production, induction of pathogen resistance, nutrient solubilization, mycorrhizal functioning, and decreased pollutant toxicity, all contribute to the promotion of plant growth by PGPR [14]. When resources are scarce [15], phosphorus solubilization, in particular, might result in the creation of biofertilizers [16].

*Bacillus* and *Pseudomonas* are the two main genera of PGPR. According to [17], the ability of strains of the endospore-forming *B. amyloliquefaciens* to colonize plant rhizospheres, promote plant growth, and reduce competing phytopathogenic bacteria and fungi sets them apart from other representatives of the genus. *B. subtilis* is a typical plant growth-promoting rhizobacterium. These bacteria have both biofertilizer and biocontrol functions, and are the most commonly used in agricultural production [18].

Cyanobacteria are one of the oldest life forms in the earth’s atmosphere [19]. The only source of biogenic oxygen and a significant supply of fixed carbon and nitrogen, they are also the ultimate ancestors of all plastids [20]. One of the most widely studied beneficial microorganisms, with numerous applications in the agricultural sector, is the cyanobacterium [21,22] Compared to non-inoculated plants, plants inoculated with this microorganism displayed higher density of smaller stomata, a thicker palisade parenchyma, and larger intercellular spaces in the mesophyll. These differences were related to changes in leaf function anatomical traits, promoting leaf gas exchange in soybean leaves grown hydroponically with plant growth-promoting microorganisms. The thicker leaf lamina of inoculated plants did not increase mesophyll resistance, compared to non-inoculated plants, because it was accompanied by increased intercellular spaces, which improved access to carboxylation sites of the chloroplasts inside the cypsela. In addition, inoculation was associated with higher photochemical efficiency in adult plants during the seed maturation stage, thanks to the higher efficiency in using and converting light within photosystems. Moreover, the increased thickness of the palisade parenchyma, which houses the majority of the chloroplasts, correlated with better photosynthesis in inoculated plants. More precisely, inoculation has been reported to be associated with characteristics such as palisade thickness and spongy parenchyma porosity, and beneficial effects on lateral and vertical gas transport inside the leaf lamina [23].

The cultivation of crops and the requirement of high-quality food for the world’s expanding population depend on plant nutrients. Therefore, plant nutrients are essential to sustainable agriculture. Recently, there has been increasing interest in the use of beneficial microorganisms as an attractive alternative to chemical inputs, which are polluting and decreasing soil fertility, to induce plant resistance to biotic and abiotic stress factors and increase plant growth and yield. The aim of this study was to evaluate the impact of PGPR in improving the production of tomato fruits through their association with tomato plant roots. This study significantly contributes to plant growth sustainability and plant maturation for fruiting. This paper also examines the application of different PGPR compositions to improve tomato plants’ environmental health status.

## 2. Results

### 2.1. Soil Analysis: Dehydrogenase Activity (DHA)

Figure 1 illustrates the influence of biofertilizer inoculation on the activities of dehydrogenase (DHA) in the rhizosphere soil of tomato plants over two seasons. As shown by the results, biofertilizer inoculation significantly increased the enzyme activity in the rhizosphere soil of the inoculated plants. The DHA enzyme activity in the second season was more sensitive than in the first season, with about 21% of its activity being lost in the second season while only 11% was lost in the first season, compared with the uninoculated treatment (control). With inoculation, the first-season DHA activity in soil of *B. subtilis* + *B. amyloliquefaciens*-treated plants surprisingly increased to the maximum activity among all treatments (243 μgTPF/g soil/day), while the activity of the same enzyme was slightly increased in the mixed inoculation with *B. subtilis* + cyanobacteria (232 μgTPF/g soil/day compared with its uninoculated counterpart, which was (98μgTPF/g soil/day). Similarly, in the second season, plants inoculated with *B. subtilis* + *B. amyloliquefaciens* mixture showed DHA activity increased by 315%, and mixed inoculation with *B. subtilis* + cyanobacteria showed the same trend, compared with the control treatment.

### 2.2. Plant Analysis

#### 2.2.1. Photosynthetic Pigments

The effect of biofertilizer’s on the photosynthetic pigments of the inoculated and un-inoculated tomato plants is presented in Figure 2A,B. As shown from the results, inoculation significantly increased pigment content. In the first season, inoculation with the *B. subtilis* + *B. amyloliquefaciens* mixture and B. amyloliquefaciens showed the highest significant effect; where the recorded increase of chlorophyll a content was 30% and 27%, respectively, in chlorophyll b it was 20% and 16%, respectively, in total chlorophyll, it was 54% and 43%, respectively, and in carotenoids it was 52% and 42%, respectively, compared with the control. Meanwhile, the second season of treatment had a more significant impact on chlorophyll a, chlorophyll b, total chlorophyll, and carotenoid contents, the content of which increased by 48%, 31%, 38%, and 35%, respectively, in plants inoculated with the *B. subtilis* + *B. amyloliquefaciens* mixture, compared with uninoculated ones.

#### 2.2.2. Anatomical Studies

Microscopic measurements were made of certain anatomical characteristics in transverse sections of tomato (*S. lycopersicum*) hybrid Desira leaflets, as compared to control plants. Microphotographs are presented in Figure 3 and Figure 4A–C. The application of *B. subtilis* + *B. amyloliquefaciens* was associated with the greatest increases in the thickness of the upper and lower epidermal layers of tomato leaflets by +49.1% and +50%, respectively. The application of *B. subtilis* + *B. amyloliquefaciens* + cyanobacteria resulted in the lowest increases in the thickness of the upper and lower epidermal layers of 5.7% and 10.9%, respectively, compared to the untreated plants. Additionally, the highest increases in palisade tissue, spongy tissue, and the length of vascular bundle, of 45.2%, 36.7%, and 47.4%, respectively, compared to control (Figure 3), were associated with the application of *B. subtilis* + *B. amyloliquefaciens* (Figure 3). However, the application of *B. subtilis* + *B. amyloliquefaciens* + cyanobacteria led to the lowest increases in palisade tissue, spongy tissue, and vascular bundle thickness, by 5.1%, 5.3%, and 3.9%, respectively, compared to the control (Figure 3). As shown in Figure 4B, it was clear that application of *B. subtilis + B. amyloliquefaciens* was associated with the highest increase in the thickness of the midrib zone, by 32.2%, compared to the control (Figure 4A), but application of *B. subtilis* + *B. amyloliquefaciens* + cyanobacteria led to slightly increased thickness of the midrib zone by 1.8% (Figure 4C) compared to the control (Figure 4A).

### 2.3. Yield Parameters

#### 2.3.1. Productivity Criteria

The influence of bacterial inoculation with two PGPR strains (*B. subtilis* and *B. amyloliquefaciens*) and cyanobacteria on the yield parameters of tomato plants is presented in Table 1. As shown by the results, in the first season, the effect of inoculation varied among the measured parameters. As shown in the results, the smallest value of plant height, 77 cm, was recorded for the uninoculated treatment, which was significantly increased to 117 cm and 109 cm after inoculation with the *B. subtilis* + *B. amyloliquefaciens* mixture and *B. amyloliquefaciens*, respectively. Similarly, the number of leaves was increased by 69% and 50%, respectively, in the treatment groups compared with their uninoculated counterparts. Additionally, the highest value for the number of branches per plant was recorded in the plants inoculated with the *B. subtilis* + *B. amyloliquefaciens* mixture, with a 2-fold increase, relative to the uninoculated group. In the second season, inoculation with the *B. subtilis* + *B. amyloliquefaciens* mixture was the most significant treatment across all parameters, compared with its uninoculated counterparts.

The data illustrated in Table 2 show the influence of bacterial inoculation with the two PGPR strains (*B. subtilis* and *B. amyloliquefaciens*) and cyanobacteria on fruit features of tomato plants in the two seasons (2020/2021) and (2021/2022). In both seasons, fruit number/plant was significantly increased in plants treated with the *B. subtilis* + *B. amyloliquefaciens* mixture and cyanobacteria by 76% and 60% and by 47% and 41% in the first and second seasons, respectively, compared with the uninoculated plants. Similar results were recorded for fruit weight (g), which increased in *B. subtilis* + *B. amyloliquefaciens* mixture-inoculated plants by 33% and 41% in the two seasons, relative to the uninoculated plants. Inoculation with the *B. subtilis* + *B. amyloliquefaciens* mixture significantly promoted fruit height and fruit diameter, which were increased by 50% relative to uninoculated plants.

#### 2.3.2. Antioxidant Compound (Ascorbic Acid)

The impact of biofertilizer inoculation with two bacillus (*B. subtilis* and *B. amyloliquefaciens*) strains and cyanobacteria on the amount of ascorbic acid (A.A.) in the fruits of tomato plants is illustrated in Figure 5. The results showed that the accumulation of A.A. was promoted by individual inoculation, as in *B. amyloliquefaciens*-inoculated plants, where it was significantly increased by 50%, while in plants inoculated with the mixture of the two bacillus strains (*B. subtilis* + *B. amyloliquefaciens*), it was significantly increased by 75%, compared with its content in the uninoculated plants.

#### 2.3.3. Total Soluble Solids (TSS) %

The effects of inoculation on total soluble solids content in the fruits of tomato plants are illustrated in Figure 5. In both seasons, bacterial inoculation was associated with a significant increase in total soluble solids content. The highest increase was recorded in *B. subtilis* + *B. amyloliquefaciens* mixture-inoculated plants, which had an increased accumulation of total soluble solids by about 26%, relative to their uninoculated counterparts.

#### 2.3.4. Fruit Chemical Content

The effect of inoculation with PGPRs (*B. subtilis* + *B. amyloliquefaciens*) and cyanobacteria on nitrogen, phosphorus, and potassium contents in the fruits of tomato plants are illustrated in Figure 6. In both seasons, inoculation increased the nitrogen content in tomato fruits, compared with the uninoculated treatment group. Relative to the uninoculated samples, cyanobacteria inoculation resulted in increased nitrogen content, while inoculation with the (*B. subtilis* + *B. amyloliquefaciens* mixture was associated with the highest increase of nitrogen, by 56% and 60%, with respect to their uninoculated counterparts, in the two seasons.

The phosphorus content in the fruits of inoculated plants in the first season increased by 50%, while in the second season, they increased by 33%, compared with the uninoculated samples. However, inoculation with the *B. subtilis* + *B. amyloliquefaciens* mixture and cyanobacteria led to a remarkable increase in the phosphorus content in the first season, by 100% and 67%, respectively, and in the second season, by 47% and 55%, respectively, compared to their uninoculated counterparts.

The potassium content of fruits in the first and second seasons were significantly increased in inoculated samples, by 44% in the first season and 43% in the second season, relative to their uninoculated counterparts. The highest value for potassium content was recorded for the first season in plants inoculated with the *B. subtilis* + *B. amyloliquefaciens* mixture, followed by the cyanobacteria-treated ones (100% and 92%, compared with their uninoculated counterparts). The same trend was exhibited in the second season.

## 3. Discussion

In agriculture, the usage of PGPR is expanding and could provide useful substitutes for synthetic fertilizers and chemicals. Microorganisms that encourage plant growth are effective microbial competitors that can stimulate plant growth by creating phytohormones, increasing the amount of nutrients available through the creation of secondary metabolites, or acting as biocontrol agents to keep plants safe from phytopathogen infection. Numerous publications have been published on PGPR and its useful functions [24,25]. There has not yet been enough experimental research to make assumptions about how PGPR might affect fruit quality.

*B. subtilis* and *B. amyloliquefaciens* strains are widely known to serve as plant growth-promoting bacteria (PGPB). Their colonization of roots is advantageous to the bacteria as well as the host plant. Approximately 30% of the fixed carbon that plants create is secreted through root exudates. Bacterial colonization of the roots provides a source of nutrients, and, in return, plants get bacterial substances and activities that promote plant growth and protect their hosts from stress. Plant-associated *B. amyloliquefaciens* strains are distinguished by their ability to colonize plant rhizospheres, stimulate plant growth, and suppress competing phytopathogenic bacteria and fungi, as reported by [17]. *B. subtilis* forms a thin biofilm on plant roots for long-term colonization of the rhizosphere. Chemotaxis is required for *B. subtilis* to locate and colonize young roots [26]. In the 1950s, the application of dried cyanobacterial biomass to soil was initiated. This process has been helpful in producing changes pertaining to soil fertility and improved crop productivity by 15–20% in field experiments. Moreover, the method is very natural and requires no extra economic input [27]. Due to their biofertilizer and biocontrol properties, PGPB are becoming increasingly important as a natural alternative to chemical pesticides and other agrochemicals [28].

Our results showed that PGPR inoculations significantly increased the measured photosynthetic pigments, growth parameters (plant height (cm), number of leaves, number of branches/plant), fruit features (number of fruits/plant, fruit weight (g), fruit height (cm), fruit diameter (cm), total soluble solids (TSS) %, ascorbic acid (mg/100 g F.W.), and fruit chemical contents (N, P, and K), compared to the values recorded for the uninoculated plants.

With regard to growth and yield, it was found that inoculation with a mixture of *B. subtilis* and *B. amyloliquefaciens* had a plant growth-promoting effect on tomato plants, with significant increases (*p* ≤ 0.05) in plant height (cm), number of leaves, and number of branches/plant. The outcomes were consistent with those reported by other authors who also described the beneficial impact of vaccinating tomato plants with *Bacillus* strains [29]. Additionally, there have been reports relating to other bacterial genera, including *Rhizobium* [30], *Pseudomonas* [31], and *Azospirillum* [32]. This tendency might be connected to the infecting strain’s synthesis of IAA-type metabolites [33]. *B. amyloliquefaciens* SQR9, a helpful bacterium, has been employed as an exogenous strain in a commercial bioorganic fertilizer for the stimulation of plant growth and the prevention of soil-borne illnesses in the field, according to [34,35]. According to [36,37], *B. amyloliquefaciens* is able to create phytohormones and antibiotics that either directly or indirectly boost plant growth, including indole-3-acetic acid (IAA) and bacillomycin D.

Additionally, this bacterium has the capacity to colonize plant roots and develop strong bonds with its hosts [36,37,38]. Our results followed the same general trend as those reported by [39], who discovered that treatment with *B. amyloliquefaciens* enhanced tomato plant output by 8–9% under nutrient-free, healthy circumstances. Although, according to [40,41], inoculation with *pseudomonad* PGPR can enhance the fresh weight of cucumber fruits, and one strain of *B. subtilis* increased cucumber yield in comparison to *Pythium-aphanidermatum*-inoculated controls, this effect is not universal. References [1,42] state that treatment of tomato seeds with *Pseudomonas* spp. bacterial suspension, in addition to spurring the germination of tomato seeds, also had an impact on the growth and yield of tomato fruit. It was also shown that soaking tomato seeds with a *Pseudomonas alcaligenes* suspension produced a significant effect on the number of tomato leaves and on tomato plant height.

Similar to the growth parameter, the combined application of *B. subtilis* and *B. amyloliquefaciens* considerably enhanced the total chlorophyll (a + b) content in the tomato leaves, indicating an increase in the photosynthetic activity of the tomato plants. These findings were consistent with those reported by [43], in which it was found that applying *B. subtilis* No. 2 and stimulife simultaneously boosted the chlorophyll content of *Licurich* and *Moldova* Cup tomato plants. In addition, ref. [44] observed that the total chlorophyll content was significantly stimulated by *P. agglomeranset* and *S. Proteamaculansen* in comparison with other treatments; these results were also similar to those reported by [45].

The results of our detailed anatomical study revealed that there were some differences in leaflet anatomy in response to the application of *B. subtilis* + *B. amyloliquefaciens*; this treatment group showed the greatest increase in the thickness of upper and lower epidermal layers, the length of vascular bundle, and the midrib zone, compared to the untreated plants, while application of *B. subtilis* + *B. amyloliquefaciens* + cyanobacteria led to the lowest increase in the thickness of the abovementioned layers compared to untreated plants. Alterations in thickness of the midrib zone thickness corresponded to changes in the thickness of both upper and lower epidermal layers, and palisade and spongy tissues. Regarding this, ref. [22] reported that changes in functional anatomical traits promoting leaf gas exchange in soybean leaves grown in hydroponics with plant growth-promoting microorganisms occurred. The inoculated plants showed higher density of smaller stomata, a thicker palisade parenchyma, and larger intercellular spaces in the mesophyll, compared to non-inoculated plants. Additionally, the enhanced photosynthesis in inoculated plants was consistent with a thicker palisade parenchyma, which houses the majority of the chloroplasts. Additionally, ref. [23] revealed that features, including palisade thickness and spongy parenchyma, as well as their porosity, have an impact on the lateral and vertical gas diffusion inside the leaf lamina in soybean leaves. Previous studies have demonstrated that key growth parameters of tomato plants, such as plant height and number of branches per plant, significantly increased with a combined inoculation of PGPR *(B. subtilis* and *B. amyloliquefaciens*). Similar results were found for fruit features, such as number of fruits/plant, fruit weight (g), fruit height (cm), fruit diameter (cm), total soluble solids (TSS) %, ascorbic acid (mg/100 g F.W.), and fruit chemical contents (N, P, and K), compared to those in uninoculated plants. These results were in accordance with those reported by [43]. The PGPR bacterium *B. subtilis* No. 2 increased the number of fruits per branch, and fruits per plant, while also increasing the average fruit weight. In addition, soaking tomato seeds with a *P. alcaligenes* suspension had a significant effect on the number of fruits per plant, fruit weight per plant, average weight per fruit unit, and fruit weight in hectare, as reported by [1]. The results showed that ascorbic acid was significantly increased in tomato fruits. These results were in accordance with those of [37], who showed, through biochemical analysis, that the levels of total carbohydrates, ascorbic acid, and organic acids were significantly increased by *B. subtilis* inoculation in tomato fruits of two cultivars.

Based on these results, inoculation significantly improved fruit quality and the TSS of tomato fruit juice. This corroborates the assertion made by [46] that PGPR and AMF can improve tomato fruit quality. This might be connected to increase in mineral content in inoculated plants. The generation of plant growth regulators at the root interface, which stimulates root development and improves root absorption of water and nutrients from the soil, has been linked to increased nutrient uptake by plants injected with plant growth-promoting bacteria [47]. In the present study, inoculation with bacterial isolates not only improved growth, but also had an important effect on plant nutrient acquisition. The levels of N, P, and K were increased when a combined inoculation of PGPR (*B. subtilis* and *B. amyloliquefaciens*) was supplied to the plants. This increase may have been caused by the nitrogen fixation and phosphate solubilization ability of the PGPR. Increased nutrient uptake associated with seed-treated plants may be the result of a higher root–shoot ratio, resulting in enhanced nutrition [48]. Reference [49] showed that the concentrations of N, P, K, Mg, Na, Cu, and Zn in fruit were dramatically increased after *P. putida* inoculation. It has been discovered that PGPR’s improvement of N and P nutrition helps to improve colonized plants’ water status [50]. This improved water intake might promote nutrient solubility in the soil pool and, thus, boost plant nutrient uptake.

## 4. Materials and Methods

The tomato (*S. lycopersicum*) hybrid Desira was the plant cultivar used in this study. Two PGPR *B. subtilis* and *B. amyloliquefaciens*, and a cyanobacteria strain were used in this study; these were obtained from the Microbiology Department of the Soil, Water, and Environment Research Institute (SWERI), Agricultural Research Center (ARC), Giza, Egypt. Strains were maintained for long-term storage at −70 °C in Nutrient Broth (NB) with 30% glycerol. The soil used in the present experiment was sandy–clay loam soil (PH 7.84). Its character and composition were examined. A mineral fertilizer (N, P, and K) was applied according to the recommendations of the Egyptian Ministry of Agriculture.

### 4.1. Field Experiment

The field experiment was carried out at the Experimental Farm of Al-Azhar University (Cairo, Egypt) during the growing seasons of 2020/2021 and 2021/2022, to evaluate the effects of the tested cyanobacteria and two PGPRPGPR strains (*B. subtilis* and *B. amyloliquefaciens*) on growth, quality, and yield of fresh fruits and tomato plants under field conditions.

Seeds were sown in a seedbed on 20 and 24 January in the first and second seasons, respectively. Seedlings were transplanted into the open field after 45 days. Plant spacing in the open field was 100 cm between rows and 40 cm between plants in a row. Each replicate consisted of five rows where each row was 6 m long, establishing an area of 30 m^2^ for each plot. Agricultural practices were applied whenever it was necessary. The experiment included three replicates arranged in a randomized complete block design. Each plot was fertilized with ammonium sulfate (400 kg/feddan), super calcium phosphate (300 kg/feddan), and potassium sulfate (200 kg/feddan). These quantities of fertilizer were added as follows: half after two weeks of seedling growth, and the remaining quantities divided into two equal parts, of which one part was added about 6 weeks after planting and the second about 9 weeks after planting.

The seedlings were inoculated with the previously mentioned PGPR strains (*B. subtilis* and *B. amyloliquefaciens*) or NB media as control. The two PGPR strains were streaked on nutrient agar (NA) plates. According to [51], a single colony was then picked to inoculate 250 mL of NB medium and left to grow overnight at 30 °C with shaking. This culture was used for inoculation.

The inoculation of tomato seedlings was repeated 5 days later by drenching the growing medium with the same PGPR suspension and cyanobacteria powder. Two PGPR strains (*B. subtilis* and *B. amyloliquefaciens*) and one cyanobacteria strain were used to establish the following eight treatments: *B. subtilis*; *B. amyloliquefaciens*; cyanobacteria; mixture of *B. subtilis* and *B. amyloliquefaciens*; mixture of *B. subtilis* and cyanobacteria; mixture of *B. amyloliquefaciens* and cyanobacteria; and mixture of *B. subtilis*, *B. amyloliquefaciens*, and cyanobacteria. These inoculated treatments were complemented with an uninoculated one in which no inoculation was applied to the tomato roots.

At the flowering stage, sample leaves were collected, washed immediately, and blotted between two sheets of filter paper for measurement of photosynthetic pigments.

At harvest, plant height (cm), number of leaves, number of branches/plant, number of fruits/plant, average fruit weight (g), fruit height (cm), fruit diameter (cm), total soluble solids (TSS) %, ascorbic acid (mg/100 g F.W.), and fruit chemical contents were measured.

#### Soil Analysis: Dehydrogenase Activity (DHA)

Dehydrogenase (EC1.1) activity in the rhizosphere soil was determined according to the method of [52]. The rhizosphere soil was sampled (2 g) and transferred to a test tube, after which a 2 mL aliquot of 0.5% 2, 3, 5 triphenyl tetrazolium chloride (T.T.C) solution dissolved in Tris buffer (pH 7.8) was added and thoroughly mixed. The soil samples were saturated and slightly submerged by the T.T.C solution. The tubes were then sealed with rubber silicon stoppers and incubated at 30 °C for 24 h in the dark. Thereafter, 10 mL of pure acetone was added to each tube and then the tubes were left for 2 h in the dark with continuous shaking to extract the resulting pink colored triphenyl formazan (T.P.F). The suspension was then filtered through Whatman filter paper No.1. The intensity of the pink color of the filtrate was measured at a wavelength of 485 nm using a UV spectrophotometer (Spekol UV-vis Light, Westburg). Concentrations of formazan were calculated from a standard curve and presented in μg TPF/g dry soil/day. A blank treatment, including all additives without a soil sample, was considered and subtracted.

### 4.2. Plant Analysis

#### 4.2.1. Photosynthetic Pigments

Photosynthetic pigments, chlorophyll a, chlorophyll b, and carotenoids were determined quantitatively by cutting disks (2.5 cm^2^/area) from the third leaf at the top of a branch, which were extracted with di-methyl formamide (D.M.F.) solution [HCON (CH3)2] and placed overnight at a cool temperature (5 °C). Chlorophyll a and b, as well as carotenoids, were measured using a Spectrophotometer Beckman Du 7400 at wavelengths 663, 647, and 470 nm, respectively. According to the equations described by [53], values (µg/g F.W.) were calculated as follows:Chl.a = 12.70 A663 − 2.79 A647. 
Chl.b = 20.76 A647 − 4.62 A663. 
Total Chls = 17.90 A647 + 8.08 A663 
Total carotenoids = 1000XA470 −3.72chl.a − 104chl.b/229. 

#### 4.2.2. Anatomical Studies

Test materials included lamina of the terminal leaflet from the developed compound leaf on the median portion of the main stem, which were taken in the second growing season of 2021/2022, 75 days from the sowing date.

Anatomical characteristics of the tomato leaflet were as follows:Upper epidermis thickness (µm);Lower epidermis thickness (µm);Palisade tissue thickness (µm);Spongy tissue thickness (µm);Length of midvein bundle (µm);Width midvein bundle (µm).

Each value was measured for five sections, with five readings taken for each. The micro-technique was carried out according to the method described by [54].

### 4.3. Yield Parameters

#### 4.3.1. Productivity Criteria

At the end of the season (about 5 months from cultivation), criteria of productivity were determined, namely plant height (cm), number of leaves, number of branches/plant, number of fruits/plant, average fruit weight (g), fruit height (cm), and fruit diameter (cm).

#### 4.3.2. Antioxidant Compound (Ascorbic Acid)

Ascorbic acid content was determined as mg/100 g fresh weight using 2;6 dichlorophenol-indophenol dye and oxalic acid [55]

#### 4.3.3. Total Soluble Solids (TSS) %

The total soluble solids (TSS) percentage was determined as a percentage using an Abbe refractometer [55].

#### 4.3.4. Fruit Chemical Content

Determination of nitrogen (N), phosphorus (P), and potassium (K) contents of tomato fruits was carried out. Fruit samples were wet-digested using a sulfuric acid and perchloric acid mixture [56], and fruit nutrients were determined in aliquots as follows. Nitrogen was determined using the Kjeldahl method [57]. Phosphorus was determined using the stannous chloride reduced molybdo-phosphoric blue color method, ref. [57]. Potassium was determined by flame photometry [57].

### 4.4. Statistical Analysis

All data were replicated three times and the presented data are the mean values. The obtained results were subjected to analysis of variance (ANOVA) (one way ANOVA completely randomized to determine the significance between treatments using Costat software (CoHort software, California, USA).

## 5. Conclusions

Finally, it was concluded that inoculation with PGPR (*B. subtilis* and *B. amyloliquefaciens*) and cyanobacteria significantly increased growth, yield, and fruit quality of tomato plants. These results suggest that inoculation with beneficial microbes (*B. subtilis* and *B. amyloliquefaciens*) enhances plant performance at a higher rate than any other treatment. The strategy of inoculation of PGPR individually, or in a mixture, represents a new biotechnological tool to improve tomato yield, and could be utilized in other crops.

## Figures and Tables

**Figure 1 plants-11-02732-f001:**
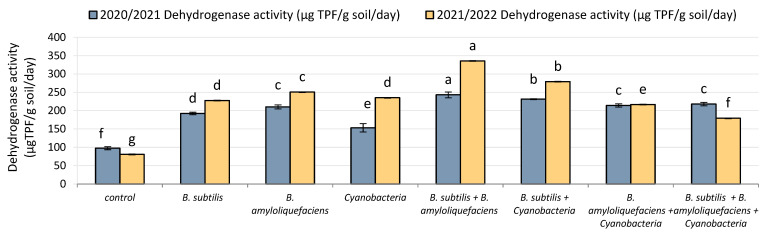
The effect of inoculation with two PGPR strains (*B. subtilis* and *B. amyloliquefaciens*) and cyanobacteria on the dehydrogenase (DHA) activity (μg TPF/g dry soil/day) in the rhizosphere soil of tomato plants in two seasons (2020/2021) and (2021/2022). Different letters indicate significant differences between treatments (ANOVA, LSD, *p* < 0.05). Error bars represent standard error of the mean.

**Figure 2 plants-11-02732-f002:**
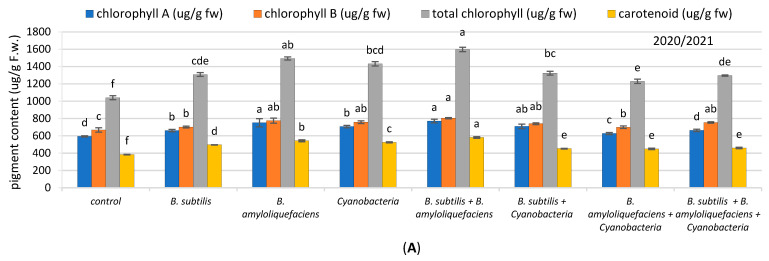
(**A**,**B**). The effect of inoculation with two PGPR strains (*B. subtilis* and *B. amyloliquefaciens*) and cyanobacteria on the pigment content (µg/g F.W.) of tomato plants in two seasons (**A**) (2020/2021) and (**B**) (2021/2022). Different letters indicate significant differences between treatments (ANOVA, LSD, *p* < 0.05). Error bars represent standard error of the mean.

**Figure 3 plants-11-02732-f003:**
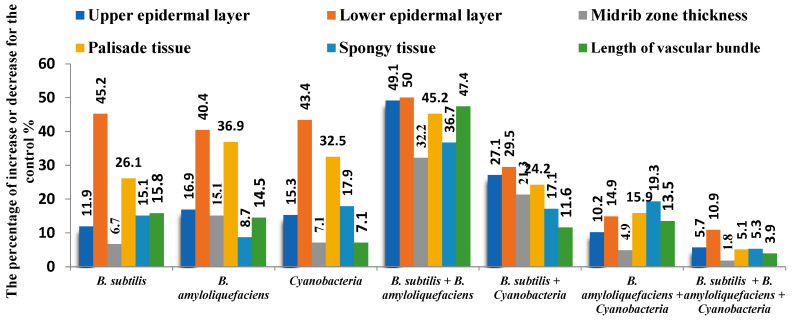
Effect of plant growth promoting rhizobacteria (PGPR) and cyanobacteria on the terminal leaflet characteristics of tomato as comparing to the untreated plants (control).

**Figure 4 plants-11-02732-f004:**
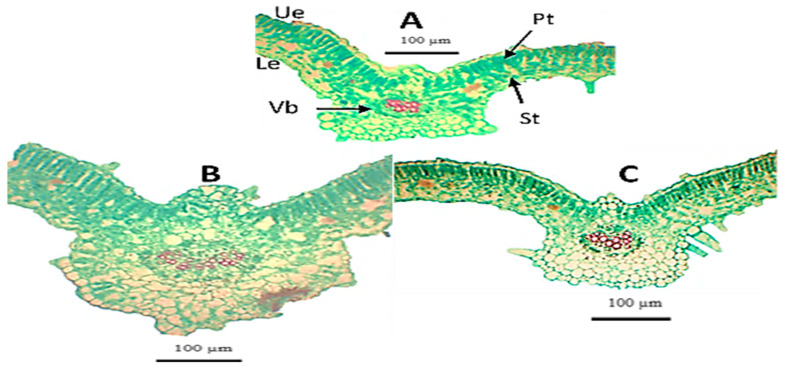
(**A**–**C**). Microphotographs of cross- sections through the blade of the terminal leaflet of the compound leaf developed on the median portion of the main stem of tomato plant. (**A**) Untreated plant (control), (**B**) from plant grown under *B. subtilis* + *B. amyloliquefaciens*, and (**C**) from plant grown under *B. subtilis* + *B. amyloliquefaciens* + *Cyanobacteria*. Abbreviations: Le = lower epidermis; Pt = palisade tissue; St = spongy tissue; Ue = upper epidermis and Vb = vascular bundle.

**Figure 5 plants-11-02732-f005:**
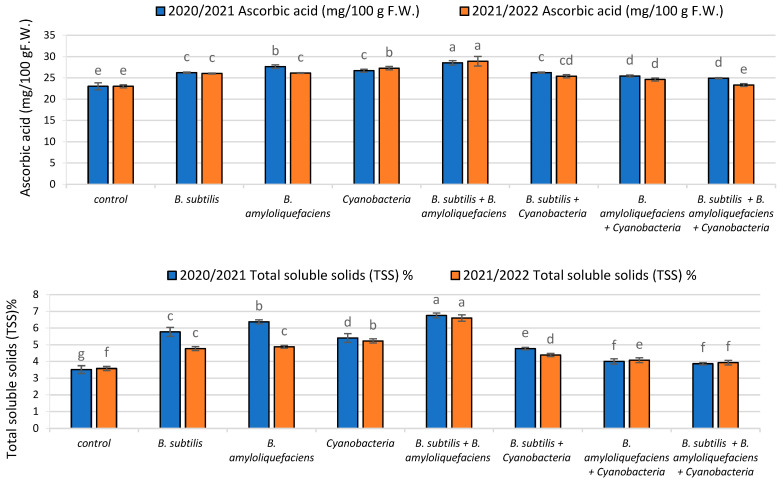
The effect of inoculation with two PGPR strains (*B. subtilis* and *B. amyloliquefaciens*) and cyanobacteria on Ascorbic acid content (mg/100g F.W.) and the Total soluble solids (TSS) % of tomato plants in two seasons (2020/2021) and (2021/2022). Different letters indicate significant differences between treatments (ANOVA, LSD, *p* < 0.05). Error bars represent standard error of the mean.

**Figure 6 plants-11-02732-f006:**
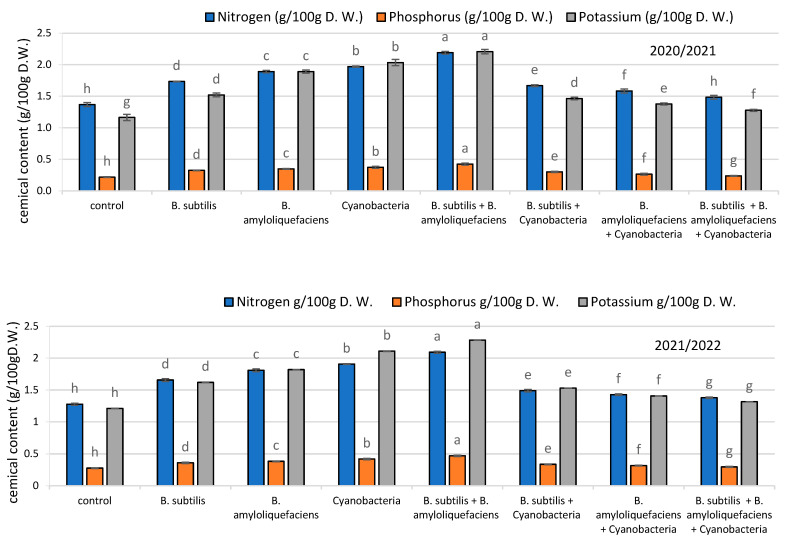
The effect of inoculation with two PGPR strains (*B. subtilis* and *B. amyloliquefaciens*) and cyanobacteria on the chemical content (Nitrogen, phosphorus and potassium) (g/100g D.w.) of tomato fruits in two seasons (2020/2021) and (2021/2022). Different letters indicate significant differences between treatments (ANOVA, LSD, *p* < 0.05). Error bars represent standard error of the mean.

**Table 1 plants-11-02732-t001:** The effect of inoculation with two PGPR strains (*B. subtilis* and *B. amyloliquefaciens*) and cyanobacteria on the yield parameters of tomato plants in two seasons (2020/2021) and (2021/2022). Different letters indicate significant differences between treatments (ANOVA, LSD, *p* < 0.05). Error bars represent standard error of the mean.

Treatment	2020/2021	2021/2022	2020/2021	2021/2022	2020/2021	2021/2022
control	77 ± 1.3 ^h^	72 ± 1.25 ^h^	36 ± 3.0 ^f^	36 ± 2.31 ^ef^	4 ± 0.58 ^d^	5 ± 0.58 ^c^
*B. subtilis*	96 ± 1.5 ^d^	93 ± 1.25 ^d^	50 ± 2.5 ^c^	47 ± 1.53 ^d^	5 ± 0.58 ^bc^	5 ± 0.58 ^c^
*B. amyloliquefaciens*	109 ± 1.4 ^b^	96 ± 0.90 ^c^	54 ± 1.2 ^b^	53 ± 1.53 ^c^	6 ± 0.58 ^ab^	5 ± 0.58 ^bc^
Cyanobacteria	104 ± 2.2 ^c^	108 ± 2.46 ^b^	53 ± 1.5 ^bc^	56 ± 0.58 ^b^	6 ± 0.58 ^ab^	5 ± 0.58 ^bc^
*B. subtilis* + *B. amyloliquefaciens*	117 ± 1.7 ^a^	117 ± 1.45 ^a^	61 ± 2.0 ^a^	64 ± 1.53 ^a^	6 ± 0.58 ^a^	6 ± 0.58 ^a^
*B. subtilis +* Cyanobacteria	87 ± 0.6 ^e^	90 ± 1.87 ^e^	45 ± 0.6 ^d^	47 ± 1.00 ^d^	5 ± 0.58 ^bc^	6 ± 0.58 ^ab^
*B. amyloliquefaciens* + Cyanobacteria	83 ± 1.6 ^f^	84 ± 1.46 ^f^	42 ± 1.0 ^e^	39 ± 1.00 ^e^	5 ± 0.00 ^bcd^	5 ± 0.58 ^bc^
*B. subtilis* + *B. amyloliquefaciens* + Cyanobacteria	81 ± 1.2 ^g^	79 ± 2.01 ^g^	39 ± 0.6 ^e^	35 ± 2.08 ^f^	5 ± 0.58 ^cd^	5 ± 0.00 ^bc^

**Table 2 plants-11-02732-t002:** The effect of inoculation with two PGPR strains (*B. subtilis* and *B. amyloliquefaciens*) and cyanobacteria on fruit features of tomato plants in two seasons (2020/2021) and (2021/2022). Different letters indicate significant differences between treatments (ANOVA, LSD, *p* < 0.05). Error bars represent standard error of the mean.

	2020/2021	2021/2022	2020/2021	2021/2022	2020/2021	2021/2022	2020/2021	2021/2022
control	25 ± 0.58 ^f^	32 ± 0.58 ^f^	91 ± 1.99 ^f^	91 ± 1.1 ^g^	4 ± 0.26 ^f^	4 ± 0.12 ^e^	4 ± 0.11 ^g^	4 ± 0.0 ^f^
*B. subtilis*	37 ± 1.15 ^c^	37 ± 1.00 ^d^	99 ± 2.11 ^d^	103 ± 1.3 ^d^	5 ± 0.13 ^cd^	5 ± 0.05 ^b^	5 ± 0.02 ^c^	5 ± 0.1 ^c^
*B. amyloliquefaciens*	39 ± 1.00 ^b^	42 ± 1.53 ^c^	105 ± 0.65 ^c^	107 ± 0.8 ^c^	6 ± 0.10 ^bc^	5 ± 0.31 ^b^	5 ± 0.05 ^b^	5 ± 0.2 ^c^
Cyanobacteria	40 ± 0.58 ^b^	45 ± 1.53 ^b^	113 ± 3.97 ^b^	122 ± 1.3 ^b^	6 ± 0.14 ^b^	6 ± 0.20 ^a^	5 ± 0.24 ^bc^	6 ± 0.1 ^b^
*B. subtilis* + *B. amyloliquefaciens*	44 ± 1.15 ^a^	47 ± 1.15 ^a^	121 ± 5.60 ^a^	128 ± 0.9 ^a^	6 ± 0.21 ^a^	6 ± 0.15 ^a^	6 ± 0.10 ^a^	6 ± 0.3 ^a^
*B. subtilis +* Cyanobacteria	37 ± 1.00 ^c^	38 ± 1.00 ^d^	98 ± 0.72 ^de^	98 ± 0.4 ^e^	5 ± 0.23 ^d^	5 ± 0.17 ^c^	5 ± 0.10 ^d^	5 ± 0.1 ^d^
*B. amyloliquefaciens* + Cyanobacteria	31 ± 1.00 ^d^	34 ± 0.58 ^e^	95 ± 0.56 ^def^	96 ± 0.8 ^e^	5 ± 0.08 ^e^	4 ± 0.22 ^d^	4 ± 0.10 ^e^	4 ± 0.1 ^e^
*B. subtilis* + *B. amyloliquefaciens* + Cyanobacteria	29 ± 0.58 ^e^	33 ± 1.00 ^ef^	94 ± 0.54 ^ef^	93 ± 1.6 ^f^	5 ± 0.26 ^f^	4 ± 0.16 ^de^	4 ± 0.03 ^f^	4 ± 0.1 ^ef^

## Data Availability

Not applicable.

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
