# Peer review of "Effects of Plant-Growth-Promoting Rhizobacteria (PGPR) and Cyanobacteria on Botanical Characteristics of Tomato (Solanum lycopersicon L.) Plants"

_plants, 2022, doi:10.3390/plants11202732_

Round 1

Reviewer 1 Report

The study conducted by Gashash et al. is well designed. Data collection and analyses are interesting. This study highlights the application of PGPR and cyanobacteria for crop improvement which has great applications in the agriculture sector.

 1)     I think there is a scope to improve the present title. Including “some” in the title is bit weird. I would suggest reframing title using scientific words.

 2)     Abstract: “tomatoes are a significant agricultural product” please reframe. After reading abstract only, I can suggest authors to get this paper checked by native speaker for English language improvement. For example, see below sentences:

        i.          tomatoes are regarded as being very beneficial to human health (grammatically incorrect)

      ii.          In this study we investigated the effect of plant growth promoting rhizobacteria (PGPR) Bacillus subtilis and B. amyloliquefaciens and cyanobacteria when utilized alone and in conjunction with each other on growth, quality and yield of fresh fruits of tomato plant. (No proper punctuation)

    iii.          spongy parenchyma as well as thickness of and changed the dimension of vascular bundles. (unclear and incomplete statement). "thickness of” what? Do authors want to say “spongy parenchyma and its thickness”?

 3)     abstract should not be descriptive for research article. Authors should also mention important results in terms of values or % for more clarity. Because abstract stands alone and readers should get all information after reading abstract only.

 4)     Overall Introduction is fine. But I would suggest authors to include all these (main objectives, significant of this study, what knowledge gaps this study is going to fill and why is this study needed at present scenario) in the last paragraph of introduction.

 5)     Materials and method section is fine.

 6)     Line no. 130: “Plant growth promoting rhizobacterial and Cyanobacteria strains:” delete this.

 7)     Line no. 135: Please mention soil pH

 8)     Line no. 141: B. amyloliquefaciens and B. subtilis should be in italics

 9)     Line no. 143: “Seeds were sown in seedbed on January 20th and 24th in the first and second seasons respectively.” What does it mean??

 10) Line no. 154-155: “The seedlings were inoculated with the previously PGPR strains or nutrient broth (NB) media as control.” Make it more clear, previous PGPR strains, write name for more clarity.

 11) Line no. 166: Please mention sample size clearly.

 12) Is it possible to provide more clear images in Figure 4?

 13) Figures 1 and 5 legend: please provide full details of different colored bar what do they represent? It is difficult to follow.

Author Response

Dear reviewer I

We would like to thank you so much  for your time and effort which have spent during the revision and hope our responses meet your high expectations, your valuable comments and constructive manor helped us to improve the overall quality of our manuscript .

We have followed your instructions point by point and hope our responses and revision  can meet now your high expectations

please find the attached file of our responses

Reviewer 2 Report

This manuscript by Ebtesam A. Gashash et al. studied the effect of three kinds of plant growth promoting rhizobacteria on tomato growth. This is a very straightforward story. The authors also proved the positive effect of these PGPR on tomato growth. But there are so many questions that should be carefully revised to make this paper publishable. I will list some of them here.

  1. First of all, the whole organization of this manuscript should be checked more carefully. The introduction is too long. I can understand that the authors want to provide any much information as they can to make the readers know what they are doing. But this one is a little too long, which may distract readers' attention. Then the subsections are a total mess. The authors used "2.x" in the method section to separate different parts, but in the results part, it goes differently. Please keep the format consistent in the whole manuscript. "Soil analysis" is labeled as number 1. The "Plant analysis" is also labeled as number 1. Then each subsection, different parts are labeled as a, b, c… The authors should at least check all of these basic things before submission.
  2. Many of the figures are unclear. For example, figure 2 is arranged into two separate parts. I think they should be two sets of results coming from two different seasons. This information should be added to the figures, just like figure 6. Actually, these two parts can be labeled as figure 2a and 2b. Also, I don't know which one is the control or uninoculated treatment in the upper part. I guess maybe the first group is. I can't find control in Figure 3. I don't suggest adding one figure legend for two figures. Please use separate figure legends for figure 3 and figure 4. There is no scale bar in figure 4, so I can't decide whether there is any difference in these three pictures.
  3. There are a lot of other minor problems. For example, the species names should be italic. Please check the whole manuscript more carefully.

Author Response

Dear reviewer II

We would like to thank you so much  for your time and effort which have spent during the revision and hope our responses meet your high expectations, your valuable comments and constructive manor helped us to improve the overall quality of our manuscript .

We have followed your instructions point by point and hope our responses and revision  can meet now your high expectations

please find the attached file of our responses

Reviewer 3 Report

Keywords: Change “PGPR, cyanobacteria, tomato” because they are in the title.

Introduction:

The authority for a Latin name should be provided after the first time it is referred to in the title, abstract, main body, and a figure or table description.

Lines 51-53: Add some current information and a link or a reference about this information.

Scientific names must write in italics. Check all text.

Line 103-108: I don't understand how these phrases relate to the previous paragraphs. They seem to be placed there randomly.

I recommend using the “Plants Microsoft Word” template file so you can adapt the paper to the standards of this journal.

When a species is cited for the first time in the text, the full text should be written for example Solanum lycopersicum L.. But afterwards it is abbreviated, as for example S. lycopersicum.

Results:

Put this section following the standards of this journal, adapting it to the standards of the journal.

All figures: Add the post hoc analysis that you have used. For example: “Different letters indicate significant differences between substrates (ANOVA, LSD, P < 0.05). Error bars represent standard error of the mean”

Figures 1, 5: Write in italics all species. Add the meaning of the colours in the description of the figure.

Figures 2, 5: Add the season in the figures and their descriptions.

Lines 263-270: Write correctly “chl.a and chl. b”

Figure 3 and Figure 4: Separate these figures and write a complete description of each one

Figure 6: Write in italics all species. Add the meaning of the colours in the description of the figure. Add the season in the figures and their descriptions. Add the units.

All results : “pgpr”, write in capital letter because in the line 88, you have written in this way.

Discussion

When a species is cited for the first time in the text, the full text should be written for example Bacillus subtilis (Ehrenberg 135) Cohn. But afterwards it is abbreviated, as for example B. subtilis.

Conclusions: They should be after material and methods section.

Material and methods:

Put this section following the standards of this journal, adapting it to the standards of the journal.

When a species is cited for the first time in the text, the full text should be written for example Solanum lycopersicum L.. But afterwards it is abbreviated, as for example S. lycopersicum.

Line 147: Correct “m2.”

Lines 149-150: What is “fed.”?

Line 179, 179: Correct “hr” for “h”

Line 180: Write “two hr” or in number or in letter, but the same in all the text. Homogenize this type of data in the all document.

Add name, class or brand of the medium, kits, equipment, country, compounds, etc. For example: potato-dextrose-agar (PDA) medium (Sigma-Aldrich, St. Louis, MO, USA).

Lines 190 and 206-211: Remove the period after cm and µm

Line 192 and 194: What are “5oc” and “MU”?

Line 233: Add the post hoc analysis that you have used.

References and ALL text: scientific names are cited in italic; these names must be different of the format of the section. Adapt the references to journal standards. The genus, species, and variety name, genes must write in italics. Check all text.

Author Response

Dear reviewer III

We would like to thank you so much  for your time and effort which have spent during the revision and hope our responses meet your high expectations, your valuable comments and constructive manor helped us to improve the overall quality of our manuscript .

We have followed your instructions point by point and hope our responses and revision  can meet now your high expectations

please find the attached file of our responses

Round 2

Reviewer 2 Report

This revised version looks better than the previous one. However, the authors kept all of the changes in the revised manuscript. This makes it unreadable at all. Please resubmit a revised version with all the accepted changes and a clean version for review. 

Even though this version is difficult to read, I still spotted some questions.

First, I didn't see the error bars in Fig1 of the 2021/2022 season results. They were presented in the first version. Second, I am confused about the statistical analysis. Did the authors compare the different treatments from the same season or did the authors compare all of the data from two seasons? For example, the results of  B.amyloliquefaciens +Cyanobacteria treatment in Fig.1 in the 2020/2021 season was labeled as "c," while in the 2021/2022 season was labeled as "e". So it looks like the same treatment in different seasons showed significant differences. But they have a very similar value. So, if the analysis is conducted based on different seasons, maybe the authors can consider putting all data from the same season together. 

Author Response

Dear reviewer II

Thank you so much for your time and effort. We have followed all your suggestions and comments .

Please find the attached file of our responses

Warmest regards

The authors

Round 3

Reviewer 2 Report

This is a revised version of the article studying the role of PGPR and cyanobacteria on the botanical characteristics of tomatoes. I think my problems have been properly addressed. And it meets the merit of plants now.